# Effect of High-Pressure Processing (HPP) on the Fatty Acid Profile of Different Sized Ragworms (*Hediste diversicolor*) Cultured in an Integrated Multi-Trophic Aquaculture (IMTA) System

**DOI:** 10.3390/molecules24244503

**Published:** 2019-12-09

**Authors:** Bruna Marques, Ana Isabel Lillebø, Maria do Rosário M. Domingues, Jorge A. Saraiva, Ricardo Calado

**Affiliations:** 1Department of Biology & CESAM & ECOMARE, University of Aveiro, Campus Universitário de Santiago, 3810-193 Aveiro, Portugal; bruna.marques@ua.pt; 2Department of Chemistry & CESAM & ECOMARE, University of Aveiro, Campus Universitário de Santiago, 3810-193 Aveiro, Portugal; mrd@ua.pt; 3Department of Chemistry & QOPNA & LAQV-REQUIMTE, University of Aveiro, Campus Universitário de Santiago, 3810-193 Aveiro, Portugal; jorgesaraiva@ua.pt

**Keywords:** integrated multi-trophic aquaculture, highly unsaturated fatty acids (HUFA), docosahexaenoic acid (DHA), biosecurity

## Abstract

Ragworms (*Hediste diversicolor*) cultured under integrated multi-trophic aquaculture (IMTA) conditions display an improved fatty acids (FA) profile than conspecifics from the wild, thus being more suitable for maturation diets of marine fish and shrimp. Nonetheless, their use may represent a potential pathway for pathogens. The objective of the present study was to determine if high-pressure processing (HPP), as an approach to safeguard microbiological safety, could promote significant shifts on the FA profiles of different sized ragworms. An analysis of similarities (ANOSIM) revealed the existence of significant differences in the FA profile and lipid quality indexes (atherogenicity (AI), thrombogenicity (TI) and polyene (PI)) of control and HPP treated ragworms of all tested sizes (small, medium and large). Saturated (SFA) and monounsaturated FA (MUFA) increased after HPP, while polyunsaturated FA (PUFA; FA with 2 or 3 double bonds) and highly unsaturated FA (HUFA; FA with ≥ 4 double bonds) decreased. The amount of docosahexaenoic acid (DHA) in polychaetes exposed to HPP decreased an average of 25%, when compared with the levels recorded in control groups. The values of PI significantly decreased after HPP, while those of AI and TI displayed a significant increase. Despite the shifts in the FA profile of ragworms exposed to HPP, these still display a superior profile to that of wild specimens, namely the presence of DHA. Therefore, HPP can be considered as a suitable approach to safeguard the biosecurity of cultured polychaetes, without compromising their nutritional value, and support the principles of circular economy through the use of IMTA.

## 1. Introduction

In line with United Nations sustainable development goal 14 (SDG 14 “*life below water*”; specifically, “*conserve and sustainably use the oceans, seas and marine resources*”) the greatest global aquaculture challenges consist in harmonizing environmental, social and economic perspectives [1]. Integrated multi-trophic aquaculture (IMTA) systems are seen as green technology solutions in aquaculture [2], which enable a more sustainable aquaculture and reduce the dependency on wild stocks. Therefore, by being more socially accepted, IMTA has the potential to foster environmental sustainability and economic growth [3,4].

The inclusion of the polychaete *Hediste diversicolor* O.F. Müller, 1776, popularly known as ragworms, in IMTA systems has shown a high potential for the bioremediation of organic-rich waste produced by super-intensive fish farms, due to its ability to feed on particulate organic matter [5,6,7,8]. Plus, it has been shown that this extractive species has the ability to selectively retain and/or biosynthesize essential fatty acids (EFA), namely highly unsaturated fatty acids (HUFAs) [5,9,10]. In general, HUFAs are essential cell membranes constituents, playing a key role in membrane fluidity, in the modulation of enzyme activity, in neural development, and regulation of stress resistance, as shown for marine finfish and shrimp [11,12]. When cultured in IMTA systems, ragworms can be regarded as a potential added value product for marine fish and shrimp broodstock, as they are commonly employed whole in maturation diets for a number of species e.g., *Penaeus monodon* (Fabricius) [13] and *Solea solea* (Linnaeus, 1758) [14]. In broodstock management, the lipid and fatty acid profile of maturation diets are paramount for a high-quality development of gonads, enhanced fecundity and fertility [11,15,16]. The lack of essential fatty acids can compromise fecundity and hatching rate and induce anomalies in larvae [11]. Specifically concerning marine fish and shrimp, docosahexaenoic acid (DHA - 22:6*n*-3), arachidonic acid (AA - 20:4*n*-6) and eicosapentaenoic acid (EPA - 20:5*n*-3) have already been identified as key essential fatty acids for a normal growth and survival of marine fish and shrimp that can be provided by polychaetes [17,18]. In more detail, DHA has a major role in the structural and functional assets of cell membranes, being involved in oogenesis and embryogenesis, and impairing larval mortality and malformations [11]. Arachidonic acid (AA) is the most important precursor of prostaglandins and also play an important role in the metabolism of lipid membrane during gametogenesis in females [19]. Eicosapentaenoic acid (EPA) is also a precursor of some other types of prostaglandins and a strong inhibitor of AA-derived eicosanoid production [20]. Therefore, it is essential to have adequate DHA/EPA and AA/EPA ratios to ensure reproductive success and enhance egg quality [20]. The profile of *H. diversicolor* in these specific fatty acids somehow explains why this polychaete species stimulates gonad maturation and spawning in marine fish and shrimp [21]. If one considers the way how polychaete meal is prepared [22,23,24], the use of IMTA-cultured ragworms to produce a premium polychaete meal may not be of concern from a biosecurity point of view. Nonetheless, while disease or foodborne outbreaks associated with ragworms are not commonly reported, farmers hypothesize that the use of whole *H. diversicolor*, either fresh or frozen, even if depurated and flash frozen, may represent a vector of diseases for broodstock fish or shrimp. Therefore, the present study aims to determine the effects of high-pressure processing (HPP) on *H. diversicolor* FA profile and experimentally verify if this technology can be used for whole ragworms preservation without compromising their nutritional value.

High-pressure processing is a non-thermal preservation technology that has rapidly become highly relevant in food industry, as it represents a physical additive-free food preservation technology [25]. The most important advantages of HPP is the ability to process food at ambient or a lower temperature, while simultaneously inactivate microorganisms and spoilage catalyzing enzymes with a minimal change of food taste and nutrient content; moreover, it also improves the recovery and bioavailability of bioactive compounds and reduces food allergenicity [22,23,24].

As the inactivation of microorganisms through HPP has already been thoroughly demonstrated, including for the most common pathogens affecting aquaculture (see Table 1), this aspect was not specifically addressed in the present manuscript.

The following null hypothesis was tested: 

H_0_: There are no significant differences in the FA profile and lipid quality indexes of fresh whole depurated small, medium and large-sized *H. diversicolor* and conspecifics exposed to HPP.

## 2. Results

### 2.1. Fatty Acid Profiles

Table 2 summarizes the results of the fatty acid content of different sized *H. diversicolor* in the control group and those exposed to HPP.

Both in control and HPP samples, a total of 23 fatty acids were recorded for the different *H. diversicolor* size classes (S, M and L). In control samples of *H. diversicolor*, EPA was the dominant FA in all size classes contributing to 25%, 24% and 22% of the total FA content presented by S, M and L, respectively. The average content of this FA was 7.53 ± 0.11 µg mg^−1^ DW, 6.32 ± 0.02 µg mg^−1^ DW and 6.68 ± 0.17 µg mg^−1^ DW for S, M and L, respectively. Palmitic acid (16:0) was the most dominant SFA, with 4.09 ± 0.06 µg mg^−1^ DW, 3.46 ± 0.18 µg mg^−1^ DW and 4.57 ± 0.11 µg mg^−1^ DW for S, M and L, respectively. Concerning MUFA, the most representative fatty acid was 20:1n-9, with 2.20 ± 0.03 µg mg^−1^ DW, 1.91 ± 0.07 µg mg^−1^ DW and 2.06 ± 0.02 µg mg^−1^ DW being recorded for S, M and L, respectively. In *H. diversicolor* exposed to HPP, EPA was also the most representative FA with 6.77 ± 0.11 µg mg^−1^ DW, 5.07 ± 0.05 µg mg^−1^ DW and 5.33 ± 0.17 µg mg^−1^ DW to S, M and L, respectively. Palmitic acid content was 4.58 ± 0.06 µg mg^−1^ DW, 4.77 ± 0.07 µg mg^−1^ DW and 6.57 ± 0.18 µg mg^−1^ DW in S, M and L, respectively. The class MUFA was characterized by the fatty acids 18:1n-5 (2.35 ± 0.05 µg mg^−1^ DW, 1.99 ± 0.02 µg mg^−1^ DW and 2.86 ± 0.08 µg mg^−1^ DW to S, M and L, respectively) and the 20:1n-9 groups (2.34 ± 0.05 µg mg^−1^ DW, 1.94 ± 0.02 µg mg^−1^ DW and 2.18 ± 0.04 µg mg^−1^ DW to S, M and L, respectively).

The ANOSIM analysis performed on the FA content of polychaete samples from the control group and those exposed to HPP revealed the existence of significant differences (R = 1, *p* = 0.003). Considering size class (S, M and L) the ANOSIM analysis also showed significant differences (R = 1; *p* = 0.001) between the size group with a strong difference within each group. In fact, in HPP treated *H. diversicolor* SFA increased from 7.01; 5.89, and 7.58 µg mg^−1^ DW to 7.81; 7.85, and 10.32 µg mg^−1^ DW for S, M and L, respectively. Regarding HUFA, there was a decrease in FA content from 11.47; 10.35, and 10.37 µg mg^−1^ DW to 10.25; 7.97, and 8.44 µg mg^−1^ DW for S, M and L, respectively, with shifts in EPA content being mostly responsible for this decrease.

The SIMPER analysis (Table 3) confirmed the trends revealed by the ANOSIM analysis, with average dissimilarities recorded between the FA content of *H. diversicolor* in the control and HPP group being as follows: 3.3% for S; 7.5% for M; and 6.9% for L. Up to 8.7% of the dissimilarities recorded between S polychaetes in the control and HPP group was explained by EPA alone. Palmitic acid explained 11.4% and 13.4% of the dissimilarities recorded between the control and HPP group for M and L, respectively.

### 2.2. Lipid Quality Indexes

Table 4 summarizes lipid quality indexes, namely the atherogenicity index (AI), thrombogenicity index (TI) and polyene index (PI), recorded for the different size classes of *H. diversicolor* in the control group and those exposed to HPP. All indexes varied significantly (*p* < 0.05) in ragworms with the same size exposed to HPP, when contrasted to control organisms. While AI significantly increased in specimens exposed to HPP, TI and PI significantly decreased.

The ANOSIM analysis also revealed significant differences (R = 1; *p* = 0.003) between the FA content of polychaetes in control versus HPP treatment, as well as between indexes (R = 0.926; *p* = 0.001) (see Figure 1).

## 3. Discussion

High-pressure processing has been used with success as a food preservation process since the 1990’s [32]. This food treatment has the potential to inactivate microorganisms and reduce microbial growth [32,33]. Moreirinha et al. [23] showed that a group of important pathogenic bacteria (e.g., *Acinetobacter*, *Aeromonas*, *Escherichia coli*, *Enterobacter*, *Klebsiella*, *Photobacterium*, *Pseudomonas aeruginosa*, *Salmonella* and *Vibrio anguillarum*), some of which infect fish, decreased to undetectable levels when treated at 300 MPa during 15 min (see Table 1). Microorganisms differ in their response to HPP treatments depending on their physiological state, temperature, time and magnitude of induced pressure [23,34]. Several authors showed with success the reduction of microbial growth in fish and fish food products through the use of HPP. Relevant examples of HHP treatments are albacore tuna (*Thunnus alalunga*) treated at 310 MPa during 6 min [31]; smoked rainbow trout fillets (*Oncorhynchus mykiss*) and fresh catfish fillets (*Silurus glanis*) treated at 400, or 600 MPa during 1 and 5 min [26]; oysters (*Crassostrea gigas*) treated at 293 MPa during 2 min [27]; coho salmon (*Oncorhynchus kisutch*) treated at 135, 170 and 200 MPa during 30 s [28]; rainbow trout (*Oncorhynchus mykiss*) and Mahi Mahi (*Coryphaena hippurus*) treated at 150, 300, 450, and 600 MPa during 15 min [30]; Atlantic salmon (*Salmo salar*) treated at 150 MPa and 300 MPa during 15 min [29] (see Table 1).

Concerning the effects of HPP on the FA content of *H. diversicolor*, the present study shows that HPP treatment at 300 MPa during 15 min had a significant effect. Even though these results differ from a study where Atlantic salmon (*S. salar*) was exposed to the same HPP treatment [29], this may be explained due to the antioxidation capacity of astaxanthin present in the salmon, recognized to be higher than that of other antioxidants [35]. However, changes in SFA content with HPP treatment are in accordance with previous results recorded for: rainbow trout (*Oncorhynchus mykiss*); mahi mahi (*C. hippurus*) [30]; Atlantic salmon (*S. salar*) [29]; and Rongchang pig [36]. SFA are not so prone to oxidation as HUFA. Indeed, HUFA are more reactive and more easily oxidized due to the double bonds that these molecules display in their carbon chain [22,37,38]. As a consequence of lipid oxidation, FA content may display more or less pronounced shifts [29,39]. These shifts can also be related to the action of heme proteins, which act as catalysts and under HPP can become denatured and more pro-oxidative [29]. The potential destruction of lipid membranes may also explain the shifts recorded [22].

In the present study it was expected that the loss of HUFA would be reflected in quality indices. Our results demonstrated that HPP treatment induced a reduction of HUFA, including AA, EPA and DHA in *H. diversicolor*. While lipid quality indexes varied significantly in polychaetes of all sizes after exposure to HPP, the nutritional quality of these organisms remained high and certainly of interest for farmers relying on these organisms to trigger maturation in marine fish and shrimp. These results are in agreement with previous findings. It has been previously shown [28] that at an exposure of 170 MPa during 30 s, and 200 MPa during 30 s, the PI values of farmed coho salmon (*O. kisutch*) display no significant differences. Another study [37] revealed that only minor differences in PI were recorded on frozen mackerel (*Scomber scombrus*) exposed to 150, 300, 450 MPa with a holding time of 0, 2.5 and 5 min. According to other study [40], the decrease in PI values after HPP indicated that the oxidation process was in progress in control samples and was stopped. In the present study, there was a decrease in PI following HPP and an increase in AI and TI values. These results are in agreement with previous works on hake (*M. merluccius*) and sardinella (*Sardinella aurita*) [40].

According to other studies, *H. diversicolor* has a high potential for bioremediation of IMTA systems, as this species is capable of retaining highly valued FA, such as HUFAS (e.g., EPA, DHA and AA) [5,10]. The amount of DHA in polychaetes exposed to HPP decreased ≈25% (from 0.20 to 0.15 mg in 100 g of total dry weight for LP; from 0.26 to 0.16 mg in 100 g of total dry weight for MP and from 0.25 to 0.21 mg in 100 g of total dry weight for SP) when compared with the control group. However, polychaetes farmed in sand filters and exposed to HPP still display a higher content of DHA than wild conspecifics (see [10]). Therefore, the present study highlights the suitability of employing HPP for the inactivation of microorganisms and reduction of microbial growth without compromising the nutritional value of ragworms.

## 4. Materials and Methods

### 4.1. Sampling and Processing of Ragworms H. Diversicolor Cultured in IMTA System

Ragworms were cultured under the IMTA set-up described by Marques et al. [8]. At the end of experimental period (5 months), each sand filter tank stocked with *H. diversicolor* was sampled (five replicates per tank) using a hand corer (Ø 110 mm, 150 mm depth). In the laboratory, all sampled specimens were sorted into three pre-established size classes (small, total length (TL) < 30 mm; medium, TL between 30 and 50 mm; and large, TL > 50 mm) and left to depurate for 24 h in aerated containers with pre-combusted sand (at 450 °C for 5 h) and artificial seawater (prepared by mixing Tropic Marin Pro Reef salt (Tropic Marin, Wartenberg, Germany) and freshwater purified by a reverse osmosis unit and matching the salinity of 21 at the IMTA facility). In order to test the effect of HPP in *H. diversicolor*, six samples of each polychaete size class were weighted to obtain a similar biomass (5 g), with half of those samples (three per size class) being used as a control group (fresh specimens) and the other half being stored in heat sealed hermetic plastic bags and exposed to HPP.

### 4.2. High-Pressure Processing Treatments

High-pressure processing treatment was performing using a hydrostatic press (high-pressure system U33, Unipress Equipment Division, Warsaw, Poland) in a pressure vessel of 35 mm diameter and 100 mm height, at room temperature (21 °C) using as pressurizing fluid a mixture of water and propylene glycol. Hermetic plastic bags with *H. diversicolor* samples were exposed to 300 MPa (3000 bar) during 15 min. Other study [23] reported that such HPP conditions were sufficient to inactivate pathogenic organisms (see Table 1). Following HPP treatment, all samples, including those from the control, were stored at −80 °C and then dehydrated in a freeze dryer during 24 h. Freeze dried sub-samples were mechanically homogenized and stored at −80 °C for posterior biochemical analysis.

### 4.3. Fatty Acids Analysis

The separation of fatty acids methyl esters (FAMEs) was performed using a 7890B gas chromatograph system (Agilent, Santa Clara, CA, USA) with a flame ionization detector (GC-FID) following the methodology described by Aued-Pimentel et al. [41]. The main advantage of this method is that it can be done at room temperature, thereby it reduces the risks of FA decomposition. Previous to analysis all freeze-dried samples were powdered and homogenized, being weighted accurately in a sovirel/pyrex glass tube (~50 mg of *H. diversicolor* biomass) and dissolved in 1 mL of the internal standard solution of a fatty acid 21:0 in *n*-hexane (0.021 g L^−1^). Afterwards, 0.2 mL of methalonic KOH solution (2 mol L*^−^*^1^) was added, the tube was sealed and mixed vigorously in a vortex shaker for 2 min. Following this procedure, 2 mL of a saturated NaCl solution was added, and centrifugation of the mixture took place for 5 min at 3000 rpm to separate the organic phase. Then 1 mL of organic phase was transferred into a vial and the excess of solvent was evaporated with a stream of nitrogen gas. The dried sample obtained was dissolved in n-hexane (1 mL) and analysed using a GC-FID. The detector and injector were kept at 250 °C, with hydrogen as carrier gas. Fatty acid methyl ester (FAME) were separated in a fused-silica capillary column, DB-FFAP column (30 m × 320 µm × 0.25 µm) (Agilent 123-3232; Santa Clara, CA, USA) with the following temperature programme: 75 °C (initial), 20 °C min^−1^ to 155 °C (4 min), 2 °C min^−1^ to 180 °C (16.5 min), 4 °C min^−1^ to 250 °C (44 min). The identification of the fatty acids was done by matching the peaks with previously inject internal standards. The FA content (µg mg^−1^DW) in the analysed samples was calculated considering the relation between mass the area of fatty acids and the internal standard (21:0). In the present study PUFA are defined as all FA with 2 or 3 double bonds and highly unsaturated fatty acids (HUFA) (FA with ≥ 4 double bonds) are considered separately and not within PUFA.

### 4.4. Lipid Quality Indexes

Lipid quality indexes, namely the atherogenicity index (AI) and thrombogenicity index (TI) were determined according to Ulbricht and Southgate [42], while the polyene index (PI) was calculated according to Lubis and Buckle [43].

AI and TI can be used to assess the nutritional quality of polychaetes, being calculated through Equations (1) and (2):AI = (12:0 + 4 × 14:0 + 16:0)**/**[∑MUFAs+ PUFA *n*-6 + PUFA *n*-3](1)
TI = (14:0 + 16:0 + 18:0)**/**[ 0.5 × ∑MUFAs+ 0.5 × PUFA *n*-6 + 3 × PUFA *n*-3+PUFA *n*-3**/**PUFA *n*-6](2)

Concerning PI, this index can be used as a measure of PUFA damage, being a good proxy for lipid oxidation, and can be calculated according to Equation (3):PI = (EPA+DHA)/16:0(3)

### 4.5. Statistical Analysis

A resemblance matrix using the content of each FA on each polychaete sample was prepared using the Bray-Curtis similarity coefficient, after performing a log (x + 1) transformation to emphasize compositional similarity rather than on quantitative differences [44]. A two-way analyses of similarities (ANOSIM) was performed to test the null hypotheses. A global R statistic was calculated to determine the differences between FA content of small, medium and large *H. diversicolor* (S, M and L, respectively) in the different treatments (control and HPP), with R values close to one indicating maximum differences between groups and values near zero indicating a complete group overlay. An identical analysis was used to identify if there were any significant differences in lipid quality indexes. To evaluate the percentage that each FA contributed to the dissimilarities recorded between treatments a SIMPER (similarity percentage) analysis was also performed, with those contributing with 50% of cumulative dissimilarities being highlighted. PRIMER v6 with the PERMANOVA+ add-on software (PRIMER-e, Auckland, New Zealand) was used to perform all statistical analysis referred above, which are described in detail in [45]. The existence of significant differences in lipid quality indexes was determined by performing a two-way analysis of variance (ANOVA) for each index, with treatment (control vs. HPP) and polychaete size (small vs. medium vs. large) being used as fixed factors, after checking assumptions. Post hoc Tukey HSD test was used whenever ANOVA results revealed significant differences (*p* < 0.05), with STATISTICA v8 software (StatSoft Inc., Tulsa, OK, USA) being used to perform these analyses.

## 5. Conclusions

From a biosecurity perspective, HPP can be considered as a suitable approach to treat ragworms and safeguard that these do not act as a pathway for pathogens when fed to valuable fish and shrimp broodstock. When such ragworms are produced under an IMTA framework and display a higher percentage of valuable HUFAs than conspecifics from the wild, there is still a positive trade-off between using this premium polychaetes and sacrificing part of their HUFA content (including EPA, DHA and AA) due to HPP to secure microbiological safety. Overall, the basis for a circular economy is supported using the present approach and contributes to SDG 14 targets concerning aquaculture, as it integrates environmental sustainability, safety and economic growth.

## Figures and Tables

**Figure 1 molecules-24-04503-f001:**
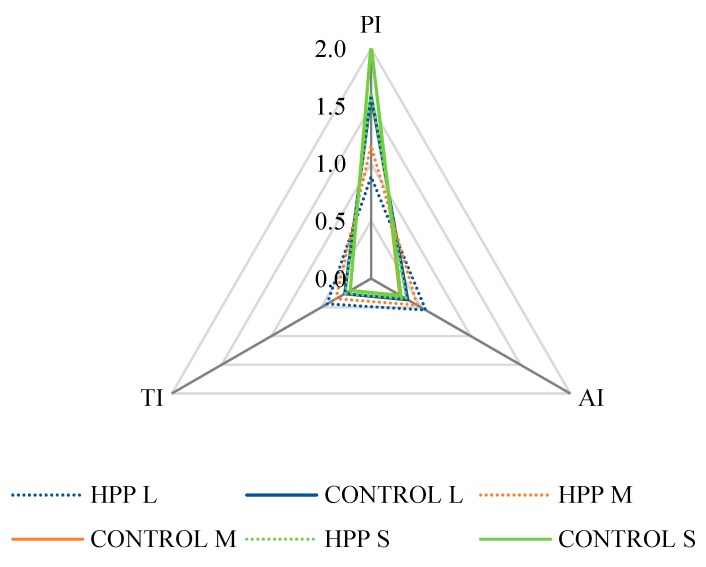
Nutritional quality indices of different sized [small (S), total length (TL) < 30 mm; medium (M), TL between 30 and 50 mm; and large (L), TL > 50 mm] cultured *Hediste diversicolor* fresh (CONTROL) or exposed to high-pressure processing (HPP). AI—Atherogenicity index; TI—Thrombogenicity index and PI—Polyene index.

**Table 1 molecules-24-04503-t001:** High Pressure Processing (HPP) conditions applied for inactivation of common pathogens in aquaculture.

Pathogens	Isolated From:	Pressure Applied/ Duration of HPP	References
*Photobacterium damselae*	Hake (*Merluccius merluccius*) and dried salted cod (*Gadus morhua*)	300 MPa (15 min)	[23]
*Vibrio anguillarum*			
*Aeromonas*			
*Salmonella sp.*			
*Escherichia coli*			
*Listeria monocytogenes*	Smoked rainbow trout fillets (*Oncorhynchus mykiss*) and fresh catfish fillets (*Silurus glanis*)	400/600 MPa (1/5 min)	[26]
*Escherichia coli*			
*Vibrio parahaemolyticus*	Oysters (*Crassostrea gigas*)	293 MPa (2 min)	[27]
*Pseudomonas spp.*	Coho salmon (*Oncorhynchus kisutch*)	135/170/200 MPa (30 s)	[28]
*Shewanella spp.*			
ND ^†^	Atlantic salmon (*Salmo salar*)	150/300 MPa (15 min)	[29]
ND ^†^	Rainbow trout (*Oncorhynchus mykiss*) and Mahi Mahi (*Coryphaena hippurus*)	150/300/450/600 MPa (15 min)	[30]
*Psycrophiles*	Albacore tuna (*Thunnus alalunga*)	310 MPa (6 min)	[31]

^†^ the authors do not define the microorganisms.

**Table 2 molecules-24-04503-t002:** Fatty acid profiles (µg mg^−1^ DW) of different sized (small, total length (TL) < 30 mm; medium, TL between 30 and 50 mm; and large, TL > 50 mm) cultured *Hediste diversicolor* fresh (Control) or exposed to high-pressure processing (HPP). Values are average of three replicates ± standard deviation.

FA	Control Small	Control Medium	Control Large	HPP Small	HPP Medium	HPP Large
**14:0**	0.29 ± 0.01	0.24 ± 0.01	0.35 ± 0.00	0.35 ± 0.00	0.41 ± 0.02	0.54 ± 0.02
**15:0**	0.25 ± 0.00	0.21 ± 0.01	0.25 ± 0.00	0.27 ± 0.00	0.28 ± 0.00	0.33 ± 0.02
**16:0**	4.09 ± 0.06	3.46 ± 0.18	4.57 ± 0.11	4.58 ± 0.06	4.77 ± 0.07	6.57 ± 0.18
**17:0**	0.45 ± 0.01	0.35 ± 0.02	0.42 ± 0.01	0.51 ± 0.01	0.44 ± 0.01	0.55 ± 0.02
**18:0**	1.92 ± 0.02	1.63 ± 0.08	1.98 ± 0.03	2.09 ± 0.03	1.95 ± 0.02	2.34 ± 0.05
**∑ SFA ^1^**	**7.01 ± 0.10**	**5.89 ± 0.30**	**7.58 ± 0.16**	**7.81 ± 0.11**	**7.85 ± 0.12**	**10.32 ± 0.29**
**16:1*n*-7**	0.78 ± 0.01	0.72 ± 0.02	0.98 ± 0.04	0.93 ± 0.01	1.00 ± 0.01	1.44 ± 0.03
**18:1*n*-9**	1.93 ± 0.03	1.65 ± 0.04	1.84 ± 0.02	2.04 ± 0.05	1.85 ± 0.02	1.88 ± 0.07
**18:1*n*-7**	1.00 ± 0.01	0.91 ± 0.03	0.99 ± 0.01	1.04 ± 0.04	1.04 ± 0.01	1.23 ± 0.06
**18:1*n*-5**	2.08 ± 0.02	1.69 ± 0.03	1.98 ± 0.06	2.35 ± 0.05	1.99 ± 0.02	2.86 ± 0.08
**20:1*n*-9**	2.20 ± 0.03	1.91 ± 0.07	2.06 ± 0.02	2.34 ± 0.05	1.94 ± 0.02	2.18 ± 0.04
**22:1*n*-9**	0.84 ± 0.01	1.05 ± 0.04	1.07 ± 0.01	0.76 ± 0.02	0.85 ± 0.01	0.82 ± 0.02
**∑ MUFA ^2^**	**8.83 ± 0.11**	**7.92 ± 0.23**	**8.92 ± 0.15**	**9.46 ± 0.23**	**8.67 ± 0.08**	**10.55 ± 0.32**
**18:2*n*-6**	0.85 ± 0.01	0.73 ± 0.03	0.78 ± 0.01	0.81 ± 0.01	0.64 ± 0.01	0.69 ± 0.03
**18:3*n*-6**	0.25 ± 0.00	0.24 ± 0.01	0.32 ± 0.02	0.25 ± 0.00	0.22 ± 0.00	0.30 ± 0.02
**20:2*n*-6**	0.63 ± 0.01	0.61 ± 0.02	0.44 ± 0.00	0.64 ± 0.01	0.56 ± 0.00	0.82 ± 0.02
**∑ PUFA ^3^**	**2.11 ± 0.02**	**1.99 ± 0.07**	**2.32 ± 0.03**	**2.04 ± 0.03**	**1.73 ± 0.10**	**2.24 ± 0.10**
**20:4*n*-6 (AA)**	0.79 ± 0.01	0.83 ± 0.04	0.82 ± 0.00	0.68 ± 0.01	0.63 ± 0.01	0.67 ± 0.03
**20:5*n*-3 (EPA)**	7.53 ± 0.11	6.32 ± 0.02	6.68 ± 0.17	6.77 ± 0.11	5.07 ± 0.05	5.33 ± 0.17
**22:4*n*-6**	0.90 ± 0.01	1.08 ± 0.05	1.01 ± 0.01	0.78 ± 0.01	0.77 ± 0.01	0.85 ± 0.04
**22:5*n*-3**	1.49 ± 0.02	1.40 ± 0.06	1.27 ± 0.04	1.38 ± 0.03	1.04 ± 0.01	1.09 ± 0.04
**22:6*n*-3 (DHA)**	0.76 ± 0.01	0.71 ± 0.02	0.59 ± 0.01	0.63 ± 0.01	0.44 ± 0.02	0.50 ± 0.03
**∑ HUFA ^4^**	**11.47 ± 0.16**	**10.35 ± 0.16**	**10.37 ± 0.23**	**10.25 ± 0.18**	**7.97 ± 0.08**	**8.44 ± 0.30**
**∑ Others ^5^**	**0.78 ± 0.01**	**0.62 ± 0.03**	**0.65 ± 0.02**	**0.93 ± 0.01**	**0.82 ± 0.1**	**1.05 ± 0.05**
**∑ Total**	**30.20 ± 0.04**	**26.77 ± 0.79**	**29.84 ± 0.59**	**30.49 ± 0.56**	**27.03 ± 0.39**	**32.60 ± 1.05**

^1^ SFA. saturated fatty acids: 14:0; 15:0; 16:0; 17:0; 18:0. ^2^ MUFA. monounsaturated fatty acids: 16:1*n*-7; 18:1*n*-9; 18:1*n*-7; 18:1*n*-5; 20:1*n*-9; 22:1*n*-9. ^3^ PUFA. Polyunsaturated fatty acids: 18:2*n*-6; 18:3*n*-6; 20:2*n*-6; 20:2*n*-9. ^4^ HUFA. Highly unsaturated fatty acids: 20:4*n*-6; 20:5*n*-3; 22:4*n*-6; 22:5*n*-3; 22:6*n*-3. ^3.4^ PUFA are defined as all FA with 2 or 3 double bonds; in the present study HUFA (FA with ≥ 4 double bonds) are not considered within ∑PUFA. ^5^ Others Fatty acids: iso 14:0, iso 15:0, iso 16:0.

**Table 3 molecules-24-04503-t003:** SIMPER overall average dissimilarities (%) between the mean fatty acid (FA) of different sized (small, total length (TL) < 30 mm; medium, TL between 30 and 50 mm; and large, TL > 50 mm) cultured *Hediste diversicolor* fresh (Control) or exposed to high-pressure processing (HPP). Contr.%—refers to the % that each individual FA contributes to the dissimilarities recorded. Cum.%—refers to the % of cumulative dissimilarities recorded for the FA listed.

Control vs. HPP Small			Control vs. HPP Medium			Control vs. HPP Large		
FA	Contr.%	Cum.%	FA	Contr.%	Cum.%	FA	Contr.%	Cum.%
**20:5*n*-3**	8.7	8.7	**16:0**	11.43	11.43	**16:0**	13.4	13.4
**16:0**	8.53	17.23	**20:5*n*-3**	8.27	19.7	**18:1*n*-7**	11.37	24.77
**18:1*n*-7**	8.09	25.32	**22:6*n*-3**	7.65	27.35	**16:1*n*-7**	9.2	33.98
**16:1*n*-7**	7.6	32.92	**22:5*n*-3**	7.13	34.48	**20:5*n*-3**	8.49	42.47
**22:6*n*-3**	7.14	40.06	**22:2*n*-6**	7.12	41.6	***iso*** **16:0**	7.07	49.53
**22:2*n*-6**	6.12	46.18	**16:1*n*-7**	6.58	48.18			
**20:4*n*-6**	5.83	52.01	**14:0**	5.57	53.75			

**Table 4 molecules-24-04503-t004:** Lipid quality indexes of different sized (small, total length (TL) < 30 mm; medium, TL between 30 and 50 mm; and large, TL > 50 mm) cultured *Hediste diversicolor* fresh (Control) or exposed to high-pressure processing (HPP). AI - Atherogenicity index; TI - Thrombogenicity index and PI - Polyene index. Different superscript letters in the same row represent significant differences at *p* < 0.05.

Lipid Quality Index	Control			HPP		
	Small	Medium	Large	Small	Medium	Large
**AI**	0.30 ± 0.00 ^a^	0.29 ± 0.02 ^a^	0.37 ± 0.00 ^b^	0.35 ± 0.00 ^b^	0.46 ± 0.01 ^c^	0.55 ± 0.00 ^d^
**TI**	0.21 ± 0.00 ^a^	0.21 ± 0.01 ^a^	0.26 ± 0.00 ^b^	0.26 ± 0.00 ^b^	0.35 ± 0.00 ^c^	0.44 ± 0.00 ^d^
**PI**	2.03 ± 0.01 ^a^	2.04 ± 0.12 ^a^	1.59 ± 0.01 ^b^	1.62 ± 0.02 ^b^	1.16 ± 0.01 ^c^	0.89 ± 0.01 ^d^

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
