# Peer review of "Effect of High-Pressure Processing (HPP) on the Fatty Acid Profile of Different Sized Ragworms (Hediste diversicolor) Cultured in an Integrated Multi-Trophic Aquaculture (IMTA) System"

_molecules, 2019, doi:10.3390/molecules24244503_

Round 1
Reviewer 1 Report
The abstract is very generaly and dont show significant results. In manuscript author mentioned pathway for pathogens. It is very big mistake that authors didnt evaluated some microorganisms in ragworms. I dont understand why is table 1 in introduction and this table again mentioned HPP conditions applied for inactivation of common pathogens in aquaculture. The chapter Results is not clear, it is results and discussion.
Author Response
Reviewer 1:
The abstract is very generaly and dont show significant results.
The abstract has been significantly re-written, as recommended by Reviewers 1 and 3, with its revised version now pinpointing the most significant differences recorded in the different treatments.
In manuscript author mentioned pathway for pathogens. It is very big mistake that authors didnt evaluated some microorganisms in ragworms. I dont understand why is table 1 in introduction and this table again mentioned HPP conditions applied for inactivation of common pathogens in aquaculture.
The authors acknowledge the criticism of Reviewer 1 but we must highlight that we clearly refer in the Introduction (lines 82-84 of the original manuscript) that “As the inactivation of microorganisms through HPP has already been thoroughly demonstrated, including for the most common pathogens in marine aquaculture (see Table 1), this aspect was not specifically addressed in the present manuscript.”. Indeed, HPP is very well established as a powerful and reliable way to inactivate microorganisms. Moreover, we stress this issue in the first sentences of our Discussion (lines 166-171 of the original manuscript) that “High-pressure processing has been used with success as a food preservation process since the 1990´s [26]. This food treatment has the potential to inactivate microorganisms and reduce microbial growth [26,27]. Moreirinha et al. [23] showed that a group of important pathogenic bacteria (e.g., Acinetobacter, Aeromonas, Escherichia coli, Enterobacter, Klebsiella, Photobacterium, Pseudomonas aeruginosa, Salmonella and Vibrio anguillarum), some of which infect fish, decreased to undetectable levels when treated at 300 MPa during 15 min (see Table 1).”
Therefore, it was the understanding of the authors that it would not be relevant to test the inactivation of potential microorganisms in the ragworms exposed to HPP in the present study, has the state of the art of HPP ensures that this approach is certainly reliable to safeguard microbiological safety. Additionally, the authors would have had to inoculate some of the pathogens referred if these were not present in IMTA cultured ragworms if they wanted to pursue this issue. This would have resulted in an unnecessary additional work load and costs, only to confirm something that is solidly documented in vailable scientific literature addressing HPP and cold pasteurization.
The presence of Table 1 in our manuscript is therefore paramount to document which microorganisms that may eventually be present as pathogens in IMTA cultured ragworms can be destroyed using HPP and safeguard biosecurity
The chapter Results is not clear, it is results and discussion.
The authors do not agree with Reviewer 1, as we consider results to be clearly detailed, in a logical and fluid way that facilitates the reading of our manuscript. A quick survey of Molecules latest articles (e.g., https://www.mdpi.com/1420-3049/24/23/4421 just published the 3rd of December 2019) allows to verify that the way manuscript sections are presented is quite flexible, being possible to showcase results as a stand-alone section or along with their discussion in a “results and discussion” section. For clarity alone, the authors would like to maintain the structure of the manuscript as it is. If the Academic Editor recommends otherwise, the authors will act as instructed.
Reviewer 2 Report
This article aims to determine the effects of high-pressure processing (HPP) on H. diversicolor FA profile, verifying if this technology can be used for whole ragworms preservation without compromising its nutritional value. Authors concluded that although the ANOSIM test revealed the existence of significant differences between FA content of control and HPP treatment, the lipid quality indexes suggest that HPP does not affect nutritional quality of FA, enabling its application for polychaetae meal pasteurization without compromising its nutritional value and supporting the principles of circular economy.
Although the article is not innovative, it contains original and interesting information. Please address any known disease or foodborne outbreak associated with ragworms.
In addition, this article would be improved if the authors clarify or revise the following:
Line 24. Revise to “were”.
Lines 27-28. Referring to Table 4, the lipid quality indexes seem to show significant difference between control and HPP treatment. May need to revise the table with statistical analysis.
Line 103. Revise to “H. diversicolor” or keep “H. diversicolor” as it is but be consistent through the manuscript.
Table 2. Need statistical analysis for at least the ones showing significant difference in the table.
Lines 134-135. Revise to “H. diversicolor”.
Line 138. Revise to “Palmitic acid (16:0)”.
Line 143. Revise to “H. diversicolor”.
Lines 154-155. Revise to “H. diversicolor”.
Figure 1 is a repeat of Table 2. Is this figure necessary? In addition, replace “,” to “.”.
Lines 160-161. Revise to “[small (S), total length (TL) <30 mm; medium 160 (M), TL between 30 and 50 mm; and large (L), TL >50 mm]”.
Line 168. Revise to “H. diversicolor”.
Line 169. Clarify how the authors arrived to the figures “3.3%, 7.5%, and 6.9%”.
Table 3. Clarify “Contr. %” and “Cum. %”
Lines 177-178. Revise to “H. diversicolor”.
Data shown in Table 4 seem to have significant difference and need statistical analysis.
Figure 2 is a repeat of Table 4. Is this figure necessary? Remove “3. Discussion”.
Line 202. Revise to “Thunnus alalunga”.
Line 206. Revise to “Salmo salar”.
Lines 222-223. Table 4 seems to show significant difference between control and HPP. Revise the sentence accordingly.
Author Response
Reviewer 2:
This article aims to determine the effects of high-pressure processing (HPP) on H. diversicolor FA profile, verifying if this technology can be used for whole ragworms preservation without compromising its nutritional value. Authors concluded that although the ANOSIM test revealed the existence of significant differences between FA content of control and HPP treatment, the lipid quality indexes suggest that HPP does not affect nutritional quality of FA, enabling its application for polychaetae meal pasteurization without compromising its nutritional value and supporting the principles of circular economy.
Although the article is not innovative, it contains original and interesting information.
The authors acknowledge the Reviewer 2 for recognizing the relevance of the present study to the ongoing effort to diversify available ingredients for aquafeeds, without compromising their biosecurity.
Please address any known disease or foodborne outbreak associated with ragworms.
To the best of the authors knowledge, there are no reported disease or foodborne outbreak associated with ragworms. However, farmers have a generalized concern that polychaetes farmed under IMTA, thus using the effluents of fish-farms, may eventually accumulate microorganisms that may introduced in the systems holding valuable fish and shrimp reproductive pairs, when these are provided fresh or frozen as a maturation feed. This generalized concern, was the rationale behind the present study. For clarity, on lines 75-78 (of the revised manuscript) it now reads: “While disease or foodborne outbreaks associated with ragworms are not commonly reported, farmers hypothesize that the use of whole H. diversicolor, either fresh or frozen, even if depurated and flash frozen, may represent a vector of diseases for broodstock fish or shrimp.”
In addition, this article would be improved if the authors clarify or revise the following:
Line 24. Revise to “were”.
The abstract has been significantly re-written, as recommended by Reviewers 1 and 3. Please refer to the revised abstract.
Lines 27-28. Referring to Table 4, the lipid quality indexes seem to show significant difference between control and HPP treatment. May need to revise the table with statistical analysis.
The abstract has been significantly re-written, as recommended by Reviewers 1 and 3. Please refer to the revised abstract.
Line 103. Revise to “H. diversicolor” or keep “H. diversicolor” as it is but be consistent through the manuscript.
For clarity, the scientific name of the species was not abbreviated in figure and table captions. This is a common practice in several scientific publications, allowing figures and tables to stand alone. If the Academic Editor recommends otherwise, the authors will act as instructed.
Table 2. Need statistical analysis for at least the ones showing significant difference in the table.
We acknowledge that is a common practice to publish this type of tables and compare each of the fatty acids (or groups of fatty acids) individually using ANOVAs (or their non-parametric equivalent). However, it is known that level at which each fatty acid is present in a given organic matrix is not independent of the levels of other fatty acids (due to catabolism and de novo synthesis). As such, this type of comparison is not as statistically accurate as the one we performed (ANOSIM), as we consider the fatty acids pool as a whole and compare it between treatments as such. This is the rationale why we have not performed the “typical” analysis comparing each fatty acid (or groups of fatty acids) that has become so “mainstream” in scientific literature. Moreover, in order to do so we would also have to correct our p-levels to account for the multiple comparisons being performed; in this specific case, if we would have used a simple correction, as that of Bonferroni, our p level would no longer be 0.05, but rather be adjusted to 0.05/25 (the number of dependent variables in the table, in this case individual fatty acids and groups) = 0.002. With our n=5, it would be very easy to miss “the big picture” and fail to perceive pronounced shifts in the pool of fatty acids (please bear in mind that the maximum dissimilarity detected was of only 7.5% between treatments). As such, the authors would keep Table 2 as it is, unless the Academic Editor recommends otherwise. If so, the authors will act as instructed.
Lines 134-135. Revise to “H. diversicolor”.
For clarity, the scientific name of the species was not abbreviated in figure and table captions. This is a common practice in several scientific publications, allowing figures and tables to stand alone. If the Academic Editor recommends otherwise, the authors will act as instructed.
Line 138. Revise to “Palmitic acid (16:0)”.
Corrected as suggested.
Line 143. Revise to “H. diversicolor”.
For clarity, the scientific name of the species was not abbreviated in figure and table captions. This is a common practice in several scientific publications, allowing figures and tables to stand alone. If the Academic Editor recommends otherwise, the authors will act as instructed.
Lines 154-155. Revise to “H. diversicolor”.
For clarity, the scientific name of the species was not abbreviated in figure and table captions. This is a common practice in several scientific publications, allowing figures and tables to stand alone. If the Academic Editor recommends otherwise, the authors will act as instructed.
Figure 1 is a repeat of Table 2. Is this figure necessary? In addition, replace “,” to “.”.
Figure 1 has been deleted in the revised version of our manuscript.
Lines 160-161. Revise to “[small (S), total length (TL) <30 mm; medium 160 (M), TL between 30 and 50 mm; and large (L), TL >50 mm]”.
Corrected as suggested.
Line 168. Revise to “H. diversicolor”.
For clarity, the scientific name of the species was not abbreviated in figure and table captions. This is a common practice in several scientific publications, allowing figures and tables to stand alone. If the Academic Editor recommends otherwise, the authors will act as instructed.
Line 169. Clarify how the authors arrived to the figures “3.3%, 7.5%, and 6.9%”.
These values are an output of the SIMPER analysis performed. This means that medium (M) and large (L) sized polychaetes differed the most in tehri FA profiles. Please note that while the differences were significant, the largest one was of only 7.5% for M sized specimens. Of these 7.5% dissimilarities, Palmitic acid (16:0) accounted for 11.43% of those dissimilarities, EPA 8.27%, DHA 7.65%... This is how SIMPER data reported in our manuscript should be interpreted.
Table 3. Clarify “Contr. %” and “Cum. %”
As requested by Reviewer 2, for clarity, we have added the folloing information to the caption of Table 3 “Contr.% - refers to the % that each individual FA contributes to the dissimilarities recorded. Cum.% - refers to the % cumulative dissimilarities recorded for the FA listed.”
Lines 177-178. Revise to “H. diversicolor”.
For clarity, the scientific name of the species was not abbreviated in figure and table captions. This is a common practice in several scientific publications, allowing figures and tables to stand alone. If the Academic Editor recommends otherwise, the authors will act as instructed.
Data shown in Table 4 seem to have significant difference and need statistical analysis.
Reviewer 2 is correct, we have added the following information to subsection 4.5 Statistical analysis “The existence of significant differences in lipid quality indexes was determined by performing a two-way analysis of variance (ANOVA) for each index, with treatment (control vs. HPP) and polychaete size (small vs. medium vs. large) being used as fixed factors, after checking assumptions. Post hoc Tukey HSD test was used whenever ANOVA results revealed significant differences (p < 0.05), with STATISTICA v8 software (StatSoft Inc., USA) being used to perform these analyses.”. Superscript letters were also added to Table 4 to account for significant differences. The following sentence was added to the Results in lines 150-152 (of the revised manuscript) “All indexes varied significantly (p<0.05) in ragworms with the same size exposed to HPP when contrasted to control organisms. While AI always significantly increased in specimens exposed to HPP, TI and PI significantly decreased.”.
Figure 2 is a repeat of Table 4. Is this figure necessary? Remove “3. Discussion”.
As already referred above in the last comment by Reviewer 1, a quick survey of Molecules latest articles (e.g., https://www.mdpi.com/1420-3049/24/23/4421 just published the 3rd of December 2019) allows to verify that the way manuscript sections are presented is quite flexible, being possible to present the sections of our manuscript as they are, namely by having Discussion as an independent section. For clarity alone, the authors would like to maintain the structure of the manuscript as it is. If the Academic Editor recommends otherwise, the authors will act as instructed.
Line 202. Revise to “Thunnus alalunga”.
Corrected as suggested.
Line 206. Revise to “Salmo salar”.
Corrected as suggested.
Lines 222-223. Table 4 seems to show significant difference between control and HPP. Revise the sentence accordingly.
Reviewer 2 is right. The sentence was revised and now in the Discussion in lines 197-199 (of the revised version of the manuscript) it reads “While lipid quality indexes varied significantly in polychaetes of all sizes after exposure to HPP, the nutritional quality of these organisms remained high and certainly of interest for farmers relying on these organisms to trigger maturation in marine fish and shrimp.”.
Reviewer 3 Report
Need to reorganize the manuscript. The authors have (1). Introduction, (2) Results, (4), Materials and Methods,and (5). Conclusions. The correct order is (1) Introduction, (2) Materials and Methods, (3) Results and Discussion, and (4) Conclusions.
Also in the Abstract Lines 25, 26 and 27, the authors write: " The ANOSIM test revealed the existence of significant differences between FA content of control and HPP treatment considering each size class." My suggestion is to delete and replace the sentence with the actual differences between the control and HPP treatments.
Lines 167-169, instead of using S, M, and L spell out the letters (i.e., Small, Medium and Large).
Author Response
Reviewer 3:
Need to reorganize the manuscript. The authors have (1). Introduction, (2) Results, (4), Materials and Methods,and (5). Conclusions. The correct order is (1) Introduction, (2) Materials and Methods, (3) Results and Discussion, and (4) Conclusions.
As already referred above in the last comment by Reviewer 1, a quick survey of Molecules latest articles (e.g., https://www.mdpi.com/1420-3049/24/23/4421 just published the 3rd of December 2019) allows to verify that the way manuscript sections are presented is quite flexible, being possible to present the sections of our manuscript as they are. For clarity alone, the authors would like to maintain the structure of the manuscript as it is. If the Academic Editor recommends otherwise, the authors will act as instructed.
Also in the Abstract Lines 25, 26 and 27, the authors write: " The ANOSIM test revealed the existence of significant differences between FA content of control and HPP treatment considering each size class." My suggestion is to delete and replace the sentence with the actual differences between the control and HPP treatments.
The abstract has been significantly re-written, as recommended by Reviewers 1 and 3, with its revised version now pinpointing the most significant differences recorded in the different treatments.
Lines 167-169, instead of using S, M, and L spell out the letters (i.e., Small, Medium and Large).
For consistency with the abbreviations used over the manuscript, the authors have decided to maintain the letters S, M and L when referring the different sizes of the polychaetes. If the Academic Editor recommends otherwise, the authors will act as instructed.
Round 2
Reviewer 1 Report
It was difficult read changes on manuscript because authors didn't marked with different color.